# Classification of mouse B cell types using surfaceome proteotype maps

Marc van Oostrum [1,2,3,4,8], Maik Müller [1,2,3,8], Fabian Klein [5], Roland Bruderer[6], Hui Zhang[7], Patrick G.A. Pedrioli[2], Lukas Reiter [6], Panagiotis Tsapogas[5], Antonius Rolink[5] & Bernd Wollscheid [1,2,3,4]*

System-wide quantification of the cell surface proteotype and identification of extracellular glycosylation sites is challenging when samples are limited. Here, we miniaturize and automate the previously described Cell Surface Capture (CSC) technology, increasing sensitivity, reproducibility and throughput. We use this technology, which we call autoCSC, to create population-specific surfaceome maps of developing mouse B cells and use targeted flow cytometry to uncover developmental cell subpopulations.

[1] Biomedical Proteomics Platform, Department of Health Sciences and Technology, ETH Zurich, 8093 Zurich, Switzerland. [2] Institute of Molecular Systems Biology at the Department of Biology, ETH Zurich, 8093 Zurich, Switzerland. [3] Swiss Institute of Bioinformatics (SIB), Lausanne, Switzerland. [4] Neuroscience Center Zurich, Zurich, Switzerland. [5] Developmental and Molecular Immunology, Department of Biomedicine, University of Basel, Basel, Switzerland. [6] Biognosys AG, Schlieren, Switzerland. [7] Department of Pathology, Johns Hopkins University, Baltimore, MD, USA. [8] These authors contributed equally: Marc van Oostrum, Maik Müller. *email: wbernd@ethz.ch

Cell types are typically identified and classified using selected cell-surface proteins as this sub-proteome reflects the maturity and functional state of a cell[1,2]. A comprehensive knowledge of the surface-exposed proteome (surfaceome) is necessary for classification of cell types and to enable linkage of distinct proteotypes to functional phenotypes. Global analysis of the surfaceome is challenging for technical reasons. High-throughput protein and RNA analyses are agnostic toward spatial information, and antibody-based technologies like flow or mass cytometry are limited in their multiplexing capabilities and by the availability of high-quality antibodies. Chemical biotinylation of surface glycoproteins has been used for systematic surfaceome interrogation by affinity purification of tagged proteins[3,4] or peptides[5–7] prior to mass spectrometry (MS) analysis, and protein-level enrichment methods that quantify peptides adjacent to a tagged glycopeptide have provided deep coverage of the plasma membrane proteome[3]. These methods, however, preclude a priori separation of enriched surface proteins from nonspecific background; instead, prior knowledge regarding surface localisation is used to filter for known plasma membrane proteins. In contrast, Cell Surface Capture (CSC) technology enables direct identification of extracellular N-glycopeptides[5]. Surface biotinylated N-glycopeptides are captured and enzymatically released by peptide:N-glycosidase F (PNGase F), which catalyses cleavage of asparagine-linked glycans. This leaves a deamidation within the NXS/T consensus sequence of formerly N-glycosylated peptides, indicating both surface localisation and glycosylation site. The specific site information comes at cost of sensitivity, however. Hence CSC experiments are performed with up to $1 \times 10^8$ cells per sample[6]. The large amounts of sample required mean that it is not practical to perform CSC on primary cells.

Here, we introduce autoCSC, an automated and miniaturised CSC technology enabling surfaceome mapping of primary cell types with limited sample availability. To demonstrate the utility of autoCSC, we create population-specific surfaceome maps of developing B cells and reveal functionally distinct developmental subpopulations of immature B cells.

## Results

**Automation and miniaturisation of cell surface capture.** Miniaturisation and automation have proven beneficial in phospho-peptide[8], hydrazide[9] and antibody-based[10] enrichment processes. We reasoned that the same principles could be applied to the biotin-streptavidin system used within the CSC workflow. Additionally, we optimised the standard CSC technology, most prominently by inclusion of a catalyst for optimal surface labeling[11,12]. A schematic of the autoCSC workflow is shown in Fig. 1a. Briefly, live cells are oxidised under mild conditions with sodium-meta-periodate to generate aldehydes on cell-surface carbohydrates and are subsequently labelled with cell-impermeable biocytin-hydrazide in presence of catalyst 5-methoxyanthranilic acid[11,12]. After cell lysis and tryptic digestion, the resulting peptides are subjected to automated processing on a tailored liquid handling robot. Repeated aspiration through filter tips containing streptavidin resin provides a confined reaction space for efficient binding of biotinylated N-glycopeptides. After extensive washing, N-glycopeptides are released by PNGase F provided in a plate heated to 37 °C. Using high-resolution MS, labelled extracellular peptides are identified by deamidated asparagines within the NXS/T glycosylation consensus sequence resulting from PNGase F cleavage. Targeted feature extraction in combination with data-independent acquisition (DIA)[13] is used to quantify surface-protein abundances across multiple conditions.

**Increased sensitivity and reproducibility of automated processing.** First, we evaluated the performance of the automated and miniaturised processing compared to the manual workflow.

We prepared CSC-labelled peptides from $\sim6 \times 10^8$ HeLa cells and distributed them over 30 samples containing about 5 mg peptide each. Twenty of these samples were processed in manual mode by two different researchers (Experimenters 1 and 2) and 10 were subjected to automated processing. Experimenters 1 and 2 identified 325 and 290 extracellular N-glycopeptides, respectively (median), whereas the robot identified 1811 N-glycopeptides, a more than five-fold increase over the manual process (Fig. 1b). Next to the five-fold increased sensitivity, we further asked whether autoCSC allows for more reproducible quantification of cell surface derived N-glycopeptides compared to the manual workflow. For this, deamidated N-glycopeptides were label-free quantified by integration of mass spectrometry monitored chromatographic traces. For every N-glycopeptide, quantitative values across 10 replicates were used to calculate a coefficient of variation (CV) for autoCSC, Experimenter 1 and Experimenter 2 as measure for its quantitative reproducibility. The overall distribution of obtained peptide CVs per designated workflow are visualised in Fig. 1c. Automated processing showed the lowest variation with a median CV of 28% compared to 51% and 34% for Experimenter 1 and 2, respectively (Fig. 1c). Thus, miniaturisation and automation of the biotin-streptavidin enrichment resulted in increased sensitivity and higher quantitative reproducibility for the CSC workflow.

**Surfaceome maps of 11 commonly used cancer cell lines.** To evaluate the capability for multiplexed quantification of surfaceomes, we performed autoCSC with 11 commonly used cancer cell lines. After cell labelling and digestion, samples containing approximately 1 mg peptide were processed, and N-glycoproteins were quantified with DIA-MS. We quantified 1697 unique glycosylated asparagines located within proteotypic peptides from 900 protein groups with a median of two sites per protein group (Supplementary Fig. 1, Supplementary Data 1). Of these, 192 glycosylation sites were previously unknown based on Uniprot database annotation (Supplementary Fig. 2a). However, 84% of the annotated sites in Uniprot are based on computational prediction, thus we provide experimental evidence for 1017 Uniprot-predicted glycosylation sites. The median per cell type was 602 surface proteins, two-fold higher compared to 301 reported in the largest CSC data repository, the Cell Surface Protein Atlas[6], despite using 5- to 33-fold lower amounts of input peptides. Hierarchical clustering and principal component analysis grouped samples originating from the same cell type in close proximity (Fig. 1d, e), demonstrating that autoCSC can reliably subclassify cell types based on surfaceome information.

**Mapping the surfaceome of developing B cells.** We hypothesised that autoCSC technology would enable phenotyping of freshly isolated primary cell populations. B cell development consists of successive cellular stages beginning in the bone marrow and continuing in peripheral lymphoid tissues[14]. Developing B cells are particularly challenging to characterise by CSC due to limited sample availability from mice, relatively small size, and little diversity in surfaceome composition[15]. In order to probe applicability of autoCSC, we prepared a dilution series from 30 to $0.1 \times 10^6$ cells of a B lymphoma cell line and quantified N-glycoproteins. Compared to the number of surface proteins identified from a sample of $30 \times 10^6$ cells, we recovered a median of 62% with $1 \times 10^6$, although the variance increased below $5 \times 10^6$ (Supplementary Fig. 3). With this limitation in mind, we set out to de novo map and quantitatively compare the surfaceomes of developing B cells using autoCSC. From mice, we isolated nine consecutive stages of B cell populations (Fig. 2a) with a maximum of $1 \times 10^6$ cells per sample using

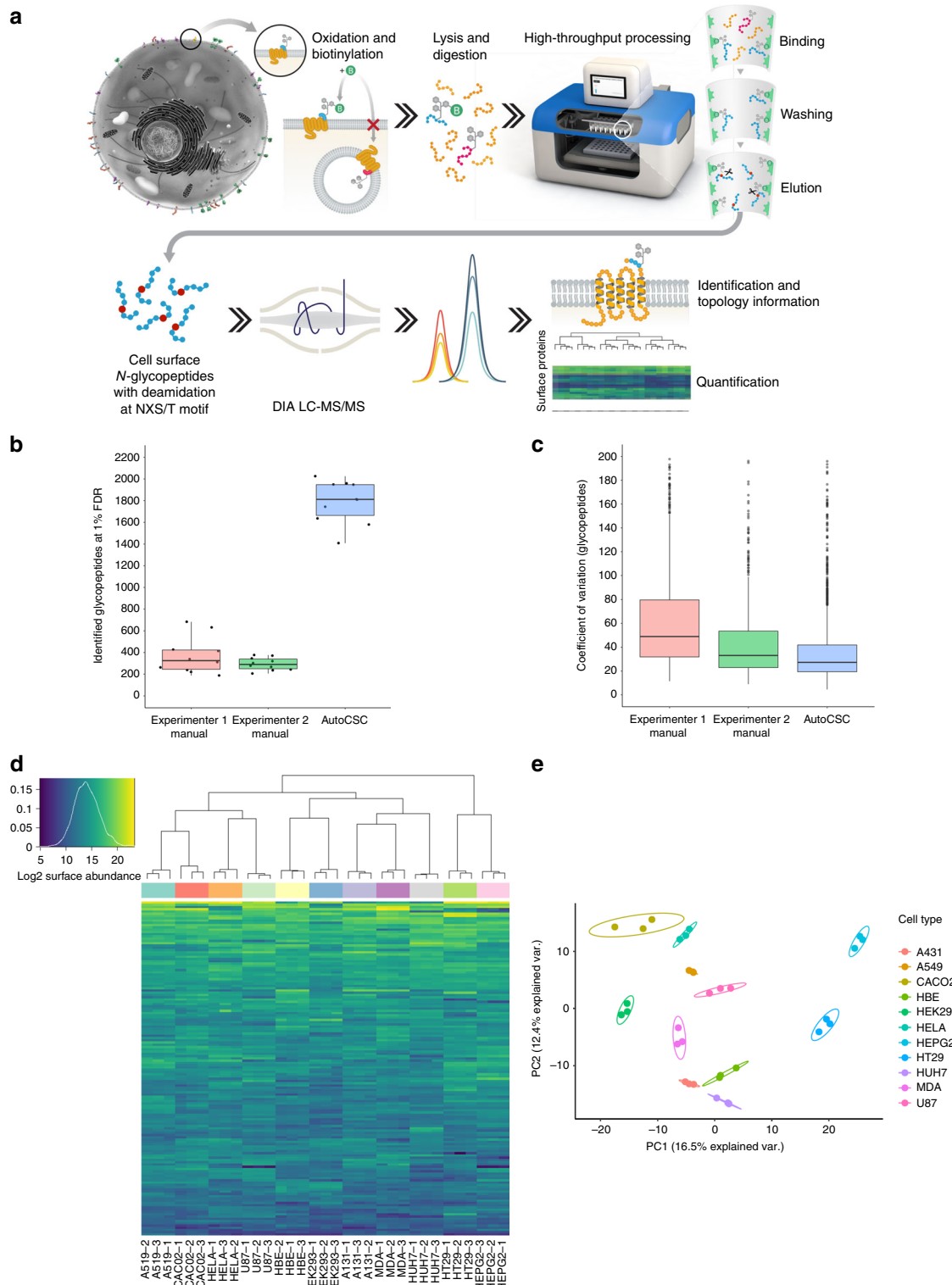

**Fig. 1 autoCSC enables quantitative surfaceome analysis with higher sensitivity, reproducibility, and throughput than the manual process. a** Schematic illustration of the autoCSC workflow. **b** Box-plots showing identified *N*-glycopeptides and **c** quantification CVs (of precursors matching glycopeptides) of manual and automated (autoCSC) processing workflows. **d** Heatmap representation and **e** principal component analysis of quantified surfaceomes in 11 common cell lines. The center line of box-plots represents the median, box limits the upper and lower quartiles, whiskers the 1.5x interquartile range and dots any outlier data points.

fluorescence-activated cell sorting (FACS). With autoCSC, we quantified 248 unique glycosylated asparagines located within proteotypic peptides and grouped them into 147 protein groups with a median of two unique sites per protein group (Fig. 2b and Supplementary Data 2). Besides finding 25 new

glycosylation sites, we also provide experimental evidence for 139 computationally predicted sites (Supplementary Fig. 2b). Few proteins we found exclusively on a particular population, for example CD80 and CD130 on B1 cells (Supplementary Fig. 5) (Fig. 2b).

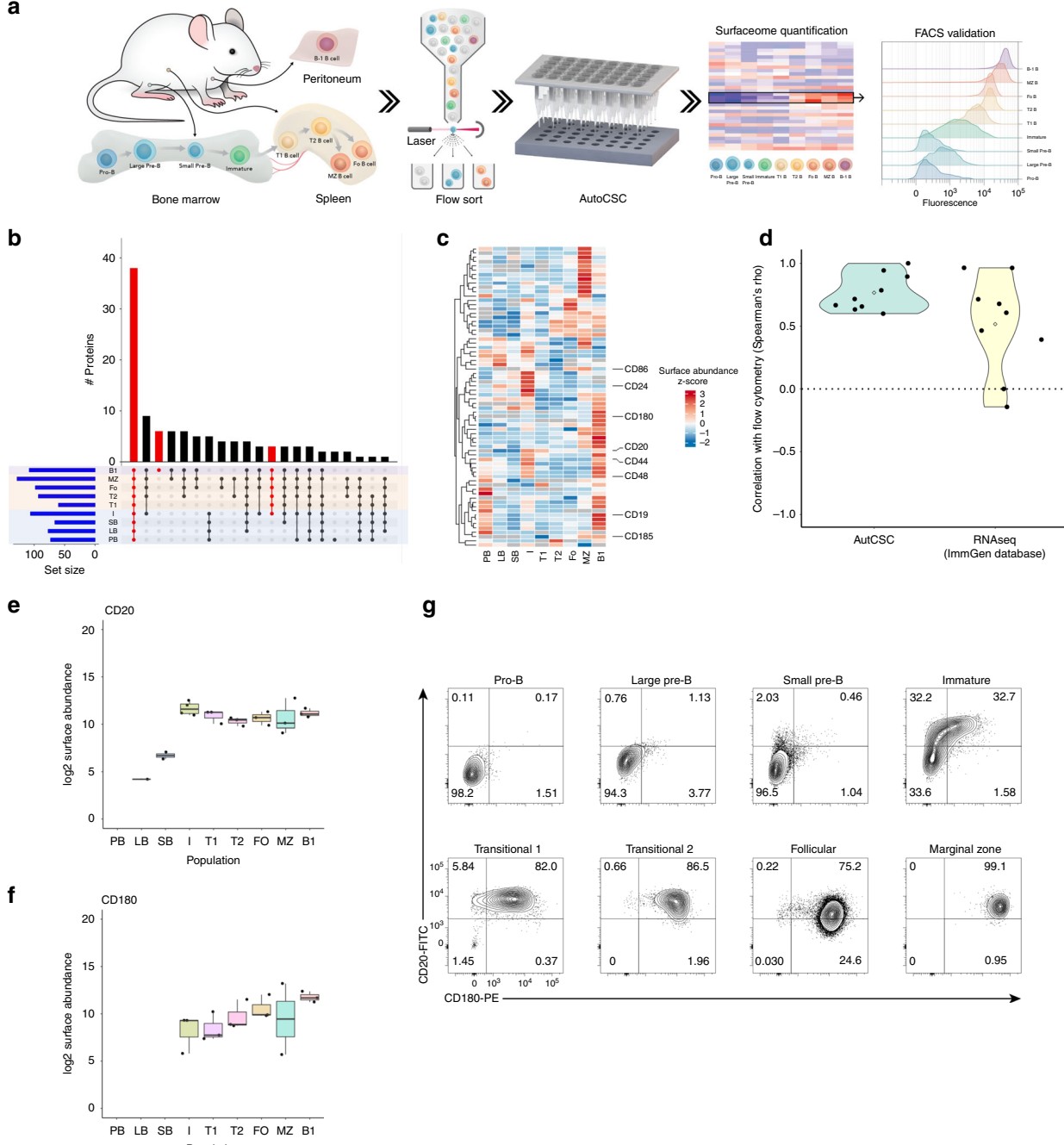

**Fig. 2 De novo mapping of developing mouse B cell populations using autoCSC. a** Schematic illustration of the workflow used to phenotype B cell populations and identify subpopulations. **b** Quantified surface protein distribution of developing B cell populations visualised by upset plot. The bar chart on top visualises the number of proteins contained within each intersection as defined in the lower part. **c** Heatmap of z-score abundance values. Surface proteins used for subsequent flow cytometry analysis are indicated to the right. **d** Correlation analysis (spearman's rho) comparing autoCSC and RNA-seq with flow cytometry. **e**, **f** Box-plots showing autoCSC obtained surface abundance values for (**e**) CD20 and (**f**) CD180 in B cell populations. The center line of box-plots represents the median, box limits the upper and lower quartiles, whiskers the 1.5x interquartile range and dots any outlier data points. **g** Flow cytometry distributions of CD20 and CD180 surface abundances during B cell development.

**Identification of immature B cell subpopulations**. Furthermore, we identified several surface proteins that were either absent or of considerably lower abundance in the bone marrow until the immature B stage, but showed stable surface abundance in later stages (Fig. 2b–f and Supplementary Fig. 5). We hypothesized that these could be used to further split the immature B stage into subpopulations with different maturities. Therefore, we performed flow cytometry analysis to identify differences in surface

abundance among individual cells within each population (Supplementary Fig. 6). First, we asked whether flow cytometry reproduced the autoCSC results for the selected proteins. All showed a strong positive correlation with an average Spearman's rho of 0.77, indicating very good agreement between the two methods (Fig. 2d). As a reference we retrieved RNA-seq data for the corresponding populations from the ImmGen database and calculated correlation coefficients with flow cytometry (Fig. 2d)[14].

Interestingly, flow cytometry revealed a bimodal distribution of CD20 within the immature B subpopulation (Supplementary Fig. 6). For CD180, we found a broad distribution covering more than two orders of magnitude (Supplementary Fig. 6). Co-staining for CD20 and CD180 across all populations revealed that prior to CD180 upregulation, CD20 abundance increased during development (Fig. 2g). Based on our data, we were able to further divide the immature B population into three subpopulations: double negative (DN), positive for CD20 but not CD180 (SP), and positive for both CD20 and CD180 (DP) (Fig. 3a). We then asked whether the three subpopulations follow a developmental trajectory towards maturity. To test the hypothesis that the DN population differentiates to DP, we cultured CD19 + IgM + CD93 + DN, SP and DP cells in vitro while monitoring CD20 and

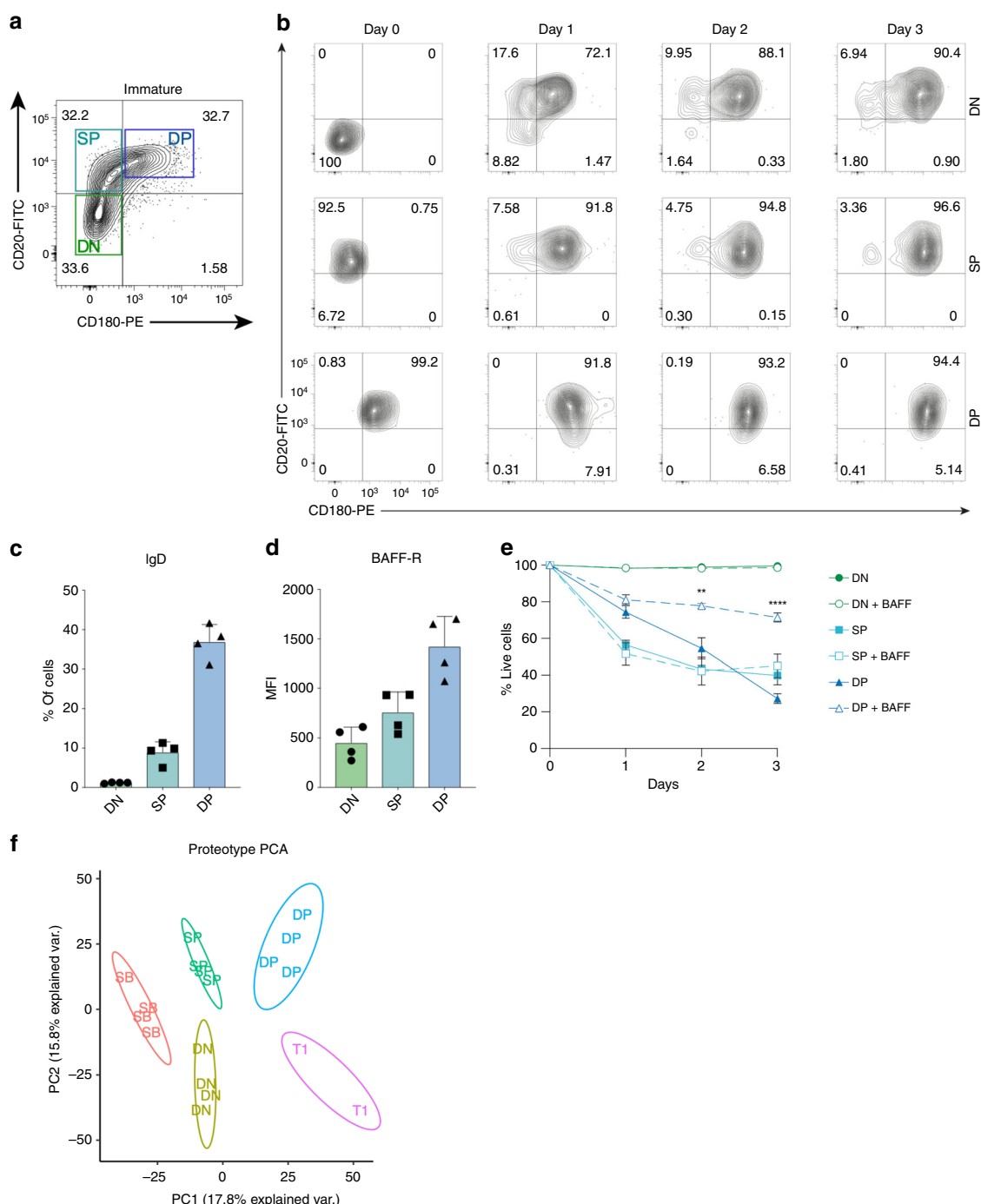

**Fig. 3 Molecular and functional assessment of immature B subpopulations. a** The immature B population was further divided into three subpopulations defined by CD20 and CD180 fluorescence intensity: double negative (DN), positive for CD20 (SP), and positive for both (DP). **b** In-vitro differentiation of the defined immature B subpopulations during three days of culturing. **c** Fraction of IgD+ cells in immature B subpopulations. **d** Mean fluorescence intensity for BAFF receptor for the three immature B subpopulations. **e** Modulation of in vitro survival of subpopulations with BAFF treatment. **f** Principal component analysis of proteotype data reflects the identified immature B subpopulations. Error bars indicate standard deviation. Statistical analysis was done with two-tailed unpaired Student's t test. One star indicates $P < 0.01$; four stars indicate $P < 0.0001$.

CD180 surface expression. In line with our hypothesis, we found that both DN and SP populations develop into DP within 3 days of culturing (Fig. 3b). Next, we probed the status of the three subpopulations with respect to known markers of maturity. We found that the DP population had a significantly higher number of IgD$^+$ cells (Fig. 3c) accompanied by higher surface abundance of BAFF receptor (Fig. 3d), as compared to SP and DN subpopulations. Based on these findings, we assessed the identified immature B subpopulations in their response to BAFF in vitro. Interestingly, BAFF treatment significantly increased in vitro survival of only the DP subpopulation (Fig. 3e), in agreement with the higher expression pattern observed for DP. For molecular assessment of the proteotypes underlying the observed phenotypes, we performed quantitative proteomic analysis of the identified immature B subpopulations (Supplementary Data 3). Also on the proteotype level, we found distinct clustering of DN, SP and DP subpopulations in a principal component analysis (Fig. 3f) and hierarchical clustering of protein quantities (Supplementary Fig. 7). Comparing all three subpopulations, we identified 435 proteins with significantly different abundance (Supplementary Data 3), including proteins involved in B-cell receptor signalling (e.g. CD79A/B, IgM) and antigen presentation (e.g. HB2A, HG2A) (Supplementary Fig. 7b, c). Finally, we probed the immature B cell subpopulations in their sensitivity towards depleting anti-CD20 antibodies. In agreement with CD20 expression levels, SP and DP subpopulations were depleted upon anti-CD20 antibody treatment in vivo, while the DN cells even slightly increased in counts (Supplementary Fig. 8a, b).

In conclusion, we identify three consecutive immature B subpopulations characterised by CD20 and CD180 surface expression and differential sensitivity towards BAFF and depleting anti-CD20 antibodies. Furthermore, we provide a comprehensive analysis of the associated proteotypes and found clustering on the proteotype-level to reflect the phenotypically defined subpopulations.

## Discussion
In summary, miniaturised and automated CSC technology was validated and used to analyse cancer cell lines and developing primary B cells. mRNA abundance is not sufficient to infer protein levels in many scenarios[16], and to predict genotype–phenotype relationships it is necessary to go beyond global protein levels and determine proteoform abundances within the spatial domains where activity is required for function[17]. CSC technology utilises an amino acid modification within a consensus sequence that reports on an initial cell-surface-restricted labelling event, enabling specific quantification with subcellular resolution. Direct identification of tagged sites has proven beneficial for spatial proteomic workflows[18] but requires the detection of a peptide bearing the modified amino acid. This caveat makes it particularly challenging to achieve identification of large numbers of formerly glycosylated sites when sample amounts are low. Insufficient sensitivity resulting at least partially from manual processing has limited CSC to specimens that can be produced in large quantities in the range of 50–100 Mio cells per sample. Physical confinement of the reaction space and automation of the biotin-streptavidin system within the autoCSC technology bridged this gap and reached the sensitivity and throughput required for broad application of surfaceome screening with primary cellular material.

These improvements will enable surfaceome proteotype maps of primary cell samples for translational research applications and systems-scale surfaceome research.

## Methods
**Chemicals.** All chemicals were purchased from Sigma unless stated otherwise.

**Mice.** C57BL/6 mice were bred and maintained in our animal facility in pathogen-free conditions. All mice used were 5–7-weeks-old. For anti-CD20 mediated depletion 100 μg IgG2b isotype control (Biolegend, Clone: RTK4530) or anti-CD20 (Biolegend, Clone: SA271G2) was injected intravenously into the tail vein 2 days before analysis.

All animal experiments were carried out under institutional guidelines (authorization number 1888 from canton Basel-Stadt veterinary office).

**Cell culture.** All cell lines were purchased from ATCC, except 16HBE14o-, which was a kind gift from Jason Mercer (MRC-Laboratory for Molecular Cell Biology, University College London). Cells were grown at 37 °C and 5% ambient $CO_2$. HepG2, HEK-293, HeLa Kyoto, Caco-2, U87, A431, MDA-MB-231, Huh-7 and A549 cells were cultured in Dulbecco's Modified Eagle's Medium, high glucose, GlutaMAX supplement (DMEM, Thermo Scientific), 10% fetal bovine serum (FBS), and 1% penicillin-streptomycin (PS). 16HBE14o- and SU-DHL-6 cells were grown in RPMI 1640 with 1.5 mM GlutaMAX, 1% PS, and 10% FBS. HT29 cells were cultured in McCoy's 5a medium with 1.5 mM GlutaMAX, 10% FBS, and 1% PS. For primary cell cultures, sorted CD19 + IgM + CD93 + DN, SP and DP cells were cultured in IMDM medium supplemented with $5 \times 10^{-5}$ M β-mercaptoethanol, 1 mM glutamine, 0.03% (wt/vol) primatone, 100 U/mL penicillin, 100 μg/mL streptomycin and 5% FBS. The effect of BAFF on the survival of cells was assessed by the addition of recombinant Fc-BAFF at a concentration of 50 ng/ml[19].

**Antibodies, flow cytometry and sorting.** Cells were flushed from femurs and tibias of the two hind legs and from the peritoneum of the mice or single-cell suspensions of spleen cells were made. Staining was performed in PBS containing 0.5% BSA and 5 mM EDTA. The following antibodies were used for flow cytometry (from BD Biosciences, eBioscience, BioLegend, or produced in house): anti-CD117 (2B8, 1/400, 562609, BD Pharmingen), anti-CD19 (1D3, 1/400, 563333, Biolegend), anti-CD127 (SB/199, 1/100, 25-1273-82, Invitrogen), anti-IgM (M41), anti-IgD (1.19, 1/1000, in-house), anti-CD93 (PB493, 1/400, in-house), anti-CD11b (M1.7015, 1/1000, in-house), anti-CD23 (B3B4, 1/400, in-house), anti-CD44 (IM7, 1/1000, in-house), anti-CD48 (HM48-1, 1/400, 11-0481-82, Invitrogen), anti-CD24 (M1/69, 1/1000, 101823, Biolegend), anti-CD20 (SA275A11, 1/400, 150407, Biolegend), anti-CD180 (RP/14, 1/400, 117706, Biolegend), anti-CD150 (TC15-12F12.2, 1/400, 115904, Biolegend), anti-CXCR5 (2G8, 1/200, 560528, BD Pharmingen), anti-PD-L1 (10 F.9G2, 1/400, 124307, Biolegend), anti-CD130 (4H1B35, 1/400, 149403, Biolegend), anti-CD80 (16-10A1, 1/400, 553768, BD Pharmingen), anti-CD86 (GL1, 1/400, 553691, BD Pharmingen), an anti-mBAFF-R (9B9, 1/400, in-house). The following FACS strategy for mouse B cell populations was used: bone marrow: proB, CD19$^+$IgM$^-$CD117$^+$; large-preB, CD19$^+$IgM$^-$CD117$^-$CD127$^+$FSC$^{large}$; small-preB, CD19$^+$IgM$^-$CD117$^-$CD127$^-$FSC$^{small}$; and immature B, CD19$^+$IgM$^+$CD93$^+$. From the peritoneum: B1, CD19$^+$CD23$^-$CD11b$^+$. From the spleen: Transitional-1, CD19$^+$CD93$^+$CD23$^-$CD21$^-$; Transitional-2, CD19$^+$CD93$^+$CD23$^+$CD21$^+$; Follicular (Fo), CD19$^+$CD93$^-$CD23$^+$CD21$^{low}$; and Marginal Zone (MZ), CD19$^+$CD93$^-$CD23$^{low}$CD21$^+$.

For flow cytometry a BD LSRFortessa (BD Biosciences) was used, and data were analysed using FlowJo Software (Treestar). For cell sorting, a FACSAria IIu (BD Biosciences) was used (>98% purity). For autoCSC, sorting was performed on four different days generating four biological replicates. Two technical replicates with $1 \times 10^6$ cells per biological replicate were processed for B cell populations (SB, I, FO, MZ) with sorting yields of $2 \times 10^6$ cells.

**Cell surface capture—labeling and digestion.** Surface glycoproteins on live cells were gently oxidised with 2 mM $NaIO_4$ (20 min, 4 °C) in labelling Buffer (LB) consisting of phosphate-buffered saline, pH 6.5. Cells were washed once in LB and subsequently biotinylated in LB containing 5 mM biocytin hydrazide (Pitsch Nucleic Acids, Switzerland) and 5 mM 5-methoxyanthranilic acid for 1 h at 4 °C min. Cells were washed three times with LB and harvested, lysed in lysis buffer (100 mM Tris, 1% sodium deoxycholate, 10 mM TCEP, 15 mM 2-chloroacetamide, pH 8.5) by repeated sonication using a VialTweeter (Hielscher Ultrasonics), and heated to 95 °C for 5 min. Proteins were digested with trypsin overnight at 37 °C using an enzyme-to-protein ratio of 1:50. For the CSC dilution series and experiments with B cells, LB used for washes contained 5% FBS, and digestion was done using LysC (Wako) and sequencing-grade trypsin (Promega), both with an enzyme-to-protein ratio of 1:200. In order to inactivate trypsin and precipitate deoxycholate, samples were boiled for 20 min, acidified with 10% formic acid to approximately pH 3, and centrifuged 10 min at 16,000 g. Peptide concentrations were determined in the supernatant using a NanoDrop 2000 instrument (Thermo Scientific). For experiments where the cell numbers were not normalised prior CSC labeling, the peptide mixtures were normalised before aliquoting into a 96-well sample plate for automated processing.

**Automated processing.** A Versette liquid handling robot (Thermo Scientific) was equipped with a Peltier element in order to adjust temperature within 96-well plates during glycopeptide elution. Streptavidin tips were prepared by pushing a bottom filter membrane into disposable automation tips (Thermo Scientific). Each tip was filled with 80 μl of Pierce Streptavidin Plus UltraLink Resin (Thermo Scientific), and tips were sealed by compressing the resin with a top filter

membrane. Assembled tips were attached to the liquid handling robot and washed with 50 mM ammonium bicarbonate by repeated cycles of aspiration and dispensing (mixing). Likewise, biotinylated peptides were bound to the streptavidin resin over 2.5 h of mixing cycles. Subsequently the streptavidin tips were sequentially washed with 5 M NaCl, StimLys Buffer (100 mM NaCl, 100 mM glycerol, 50 mM Tris, 1% Triton X-100), with 50 mM ammonium bicarbonate, 100 mM NaHCO₃, pH 11, and with 50 mM ammonium bicarbonate. For glycopeptide elution, streptavidin tips were incubated overnight in 50 mM ammonium bicarbonate containing 1000 units PNGase F (New England Biolabs) at 37 °C. All steps involving the streptavidin filter tips on the liquid handling robot are fully automated. The sample plate was then removed from the liquid handling robot and acidified to pH 2–3 with formic acid. Peptides were desalted with C18 Ultra-MicroSpin columns (The Nest Group) according to the manufacturer's instructions and dried in a SpeedVac concentrator (Thermo Scientific). For comparison of the automated with the manual workflow, manual processing was done analogously: Biotinylated peptides were bound to streptavidin in 1.5-mL tubes by addition of 80 μL washed Streptavidin Plus UltraLink Resin (Pierce), and the samples were incubated for 2.5 h on a slow rotator. Beads were washed in Mobicols (Boca Scientific) connected to a Vac-Man Laboratory Vacuum Manifold (Promega) using the same buffers. Washed beads were incubated with 50 mM ammonium bicarbonate containing 1000 units PNGase F (New England Biolabs) in a head-over-head rotator overnight at 37 °C.

**Liquid chromatography–tandem mass spectrometry (LC-MS/MS) analysis**. Samples for proteotype analysis were prepared using S-trap (Protifi) columns according to the manufacturer's instructions. For MS analysis, peptides were reconstituted in 5% acetonitrile and 0.1% formic acid containing iRT peptides (Biognosys). Depending on instrument availability and estimated required sensitivity, experiments were analysed on different LC-MS/MS systems.

The peptides resulting from the comparison between automated and manual processing were analysed in DDA mode. They were separated by reverse-phase chromatography on a high-pressure liquid chromatography (HPLC) column (75-μm inner diameter; New Objective) packed in-house with a 15-cm stationary phase ReproSil-Pur 120A C18-AQ 1.9 μm (Dr. Maisch GmbH) and connected to an EASY-nLC 1000 instrument equipped with an autosampler (Thermo Scientific). The HPLC was coupled to a Q Exactive plus mass spectrometer equipped with a nanoelectrospray ion source (Thermo Scientific). Peptides were loaded onto the column with 100% buffer A (99% H₂O, 0.1% formic acid) and eluted at a constant flow rate of 300 nL/min for 50 min with a segmented gradient from 6 to 35% buffer B (99.9% acetonitrile, 0.1% formic acid). Mass spectra were acquired in a data-dependent manner (top 12). In a standard method for medium-to-low abundance samples, high-resolution MS1 spectra were acquired at 70,000 resolution (automatic gain control target value $3 \times 10^6$) to monitor peptide ions in the mass range of 375–1500 m/z, followed by high-energy collisional dissociation (HCD)-MS/MS scans at 35,000 resolution (automatic gain control target value $1 \times 10^6$). To avoid multiple scans of dominant ions, the precursor ion masses of scanned ions were dynamically excluded from MS/MS analysis for 30 s. Single-charged ions and ions with unassigned charge states or charge states above 6 were excluded from MS/MS fragmentation.

The peptides resulting from the analysis of 11 common cell lines and the SU-DHL-6 dilution series were analysed in DIA and DDA modes for spectral library generation. For spectral library generation, a fraction of samples originating from the same cell line were pooled to generate 11 mixed pools. For the SU-DHL-6 dilution series, a pooled sample was generated for library generation by mixing fractions of the replicates with more than $10 \times 10^6$ cells input. Each sample was separated using a self-packed analytical PicoFrit column (75 μm × 50 cm length) (New Objective) packed with ReproSil-Pur 120 A C18-AQ 1.9 μm (Dr. Maisch GmbH) using an EASY-nLC 1200 (Thermo Scientific). Peptides were loaded onto the column with 100% buffer A (99% H₂O, 0.1% formic acid) and eluted at a flow rate of 250 nL/min with a segmented gradient from 1 to 52% buffer B (85% acetonitrile, 0.1% formic acid). Data were acquired on a Q Exactive HF-X mass spectrometer (Thermo Scientific). The DIA method contained 20 DIA segments of 30,000 resolution with IT set to auto, AGC of $3 \times 10^6$, and a survey scan of 120,000 resolution with 60 ms max IT and AGC of $3 \times 10^6$. The mass range was set to 350-1650 m/z. The default charge state was set to 3. Loop count 1 and normalised collision energy stepped at 25.5, 27, and 30. For the DDA, a TOP10 method was recorded with 60,000 resolution of the MS1 scan and 20 ms max IT and AGC of $3 \times 10^6$. The MS2 scan was recorded with 60,000 resolution of the MS1 scan and 110 ms max IT and AGC of $3 \times 10^6$. The covered mass range was identical to the DIA. The isolation width was set to 8 Th and the normalised collision energy to 27%.

The peptides resulting from the B cell populations were analysed in DIA and DDA modes for spectral library generation. For spectral library generation, a fraction of the samples originating from the same B cell population were pooled to generate 13 mixed pools, additionally we included 18 CSC samples from unsorted single cell suspensions from spleen (13) and bone marrow (5). Each sample was separated using an Acclaim PepMap RSLC C18, 250 mm length, 75 μm inner diameter, 2 μm particle size (Thermo Scientific), connected to a stainless steel emitter (Thermo Scientific, part. no. ES 542) using an EASY-nLC 1200 (Thermo Scientific). Peptides were loaded onto the column with 100% buffer A (98% H₂O,

2% acetonitrile, 0.15% formic acid) and eluted at a flow rate of 250 nL/min with a segmented gradient from 1 to 38% buffer B (80% acetonitrile, 0.15% formic acid). Data were acquired on a Fusion Lumos mass spectrometer (Thermo Scientific). The DIA method contained 18 DIA segments of 30,000 resolution with IT set to 50 ms and AGC of $5 \times 10^5$ and a survey scan of 60,000 resolution with 180 ms max IT and AGC of $1 \times 10^6$. The mass range was set to 350–1650 m/z. The default charge state was set to 2. Loop count 1 and normalised collision energy stepped by 2% at 27.5%. For the DDA, MS1 scans were recorded with 60,000 resolution and 50 ms max IT and AGC of $1 \times 10^6$. With a 3 s cycle time, MS2 scans were recorded with 30,000 resolution and 120 ms max IT and AGC of $2 \times 10^5$. The covered mass range was identical to the DIA. The isolation width was set to 1.4 Th, and the normalised collision energy was stepped by 2% at 27.5%.

The peptides resulting from the proteotype analysis were analysed in DIA and DDA mode for spectral library generation. For spectral library generation, a fraction of the samples originating from the same condition were pooled to generate mixed pools for each condition. Peptides were separated by reverse-phase chromatography on a high-pressure liquid chromatography (HPLC) column (75-μm inner diameter; New Objective) packed in-house with a 50-cm stationary phase ReproSil-Pur 120 A C18 1.9 μm (Dr. Maisch GmbH) and connected to an EASY-nLC 1000 instrument equipped with an autosampler (Thermo Fisher Scientific). The HPLC was coupled to a Fusion mass spectrometer equipped with a nanoelectrospray ion source (Thermo Fisher Scientific). Peptides were loaded onto the column with 100% buffer A (99% H₂O, 0.1% formic acid) and eluted with increasing buffer B (99.9% acetonitrile, 0.1% formic acid) over a nonlinear gradient for 240 min. The DIA method (Bruderer et al. 2017) contained 26 DIA segments of 30,000 resolution with IT set to 60 ms, AGC of $3 \times 10^6$, and a survey scan of 120,000 resolution with 60 ms max IT and AGC of $3 \times 10^6$. The mass range was set to 350-1650 m/z. The default charge state was set to 2. Loop count 1 and normalised collision energy was stepped at 27. For the DDA, a 3 s cycle time method was recorded with 120,000 resolution of the MS1 scan and 20 ms max IT and AGC of $1 \times 10^6$. The MS2 scan was recorded with 15,000 resolution of the MS1 scan and 120 ms max IT and AGC of $5 \times 10^4$. The covered mass range was identical to the DIA. A slightly modified acquisition method was used for the total proteotype analysis of the developmental time series experiment.

All mass spectrometric data and acquisition information were deposited to the ProteomeXchange Consortium (www.proteomexchange.org/) via the PRIDE partner repository (data set identifier: PXD013627)

**Data analysis DDA LC-MS/MS**. The peptides resulting from the comparison between automated and manual processing were analysed in DDA mode. RAW data were converted to mzML using MSconvert. Fragment ion spectra were searched with COMET (v27.0) against UniprotKB (Swiss-Prot, Homo sapiens, retrieved April 2018) containing common contaminants and MS standards. The precursor mass tolerance was set to 20 ppm. Carbamidomethylation was set as a fixed modification for cysteine, oxidation of methionine and deamidation of arginine were set as variable modifications. Probability scoring was done with PeptideProphet of the Trans-Proteomic Pipeline (v4.6.2). Peptide identifications were filtered for an FDR of ≤ 1% and presence of consensus NXS/T sequence and deamidation (+0.98 Da) at asparagines. Non-conflicting peptide feature intensities were extracted with Progenesis QI (Nonlinear Dynamics) for label-free quantification and determination of CVs.

**Data analysis DIA LC-MS/MS**. LC-MS/MS DIA runs were analysed with Spectronaut Pulsar X version 12 (Biognosys) using default settings. Briefly, a spectral library was generated from pooled samples measured in DDA (details above). The collected DDA spectra were searched against UniprotKB (Swiss-Prot, Homo sapiens or Mus musculus, retrieved April 2018) using the Sequest HT search engine within Thermo Proteome Discoverer version 2.1 (Thermo Scientific). We allowed up to two missed cleavages and semi-specific tryptic digestion. Carbamidomethylation was set as a fixed modification for cysteine, oxidation of methionine and deamidation of arginine were set as variable modifications. Monoisotopic peptide tolerance was set to 10 ppm, and fragment mass tolerance was set to 0.02 Da. The identified proteins were assessed using Percolator and filtered using the high peptide confidence setting in Protein Discoverer. Analysis results were then imported to Spectronaut Pulsar version 12 (Biognosys) for the generation of spectral libraries.

Targeted data extraction of DIA-MS acquisitions was performed with Spectronaut version 12 (Biognosys AG) with default settings using the generated spectral libraries as previously described[20,21]. The proteotypicity filter "only protein group specific" was applied. Extracted features were exported from Spectronaut using "Quantification Data Filtering" for statistical analysis with MSstats[22] (version 3.8.6) using default settings. Briefly, features were filtered for use for calculation of Protein Group Quantity as defined in Spectronaut settings, common contaminants were excluded, and presence of consensus NXS/T sequence including a deamidation (+0.98 Da) at asparagine was required for autoCSC. Features were then log-transformed, normalised, and quantified as protein abundance for each sample and condition using the quantification function in MSstats[22]. For proteotype analysis, the model estimated fold change and statistical significance for all compared conditions. Significantly different proteins were determined by the threshold |fold-change| > 1.5 and adjusted $p$-value < 0.05 of a two-sided $t$-test with the appropriate degrees of freedom.

Benjamini-Hochberg method was used to account for multiple testing. Protein abundance per sample or conditions was used for further analysis and plotting. autoCSC experiments comparing 11 cell lines and developing B cells were performed in multiple replicates (quadruplicate biological replicates and where possible additional technical replicates). Technical replicates per biological replicate were consolidated in MSstats. Outliers were removed to retain minimally three biological replicates, generating 31 samples for B cell populations and 33 samples for the 11-cell lines comparison for final quantification.

**Correlation analysis.** Proteins with abundance values (non-NA) for flow cytometry and CSC for at least four populations were considered for analysis (CD20, CD24, PDL1, CD180, CD44, CD48, CD150, CD19, CXCR5). If one condition only was not applicable it was set as zero assuming an abundance value below limit of detection. Spearman correlation coefficients were calculated and plotted using R. RNA-seq data were retrieved for the same proteins from the ImmGen database (www.immgen.org). In the retrieved dataset the following populations were sorted based on same markers as our flow cytometry analysis: Transitional 1, Transitional 2, Follicular and Marginal Zone. The following populations were considered equivalent between ImmGen and our flow cytometry analysis: Pro-B to FrB/C, immature B to FrE, and B1 to B1b. Large and small pre-B are not included in ImmGen database and were therefore not included in the correlation analysis between flow cytometry and RNA.

**Glycosylation site analysis.** For glycosylation site counting the following rules were followed: (i) only glycosylated peptides conforming to the NX[STC] consensus sequence were considered; (ii) to avoid inflating the count, non-proteotypic peptides were arbitrarily assigned to a single protein in the protein group; (iii) if a glycosylated peptide could be mapped to multiple positions within the same protein, both positions were kept, unless one of the mappings resulted in a higher number of sites matching the consensus NX[STC] motif, in which case only this one was kept. When comparing the glycosylation sites identified in this study to the ones annotated in UniProt, only proteins identified in this study as having at least one glycosylation site were considered in UniProt.

**Reporting summary.** Further information on research design is available in the Nature Research Reporting Summary linked to this article.

## Data availability

All mass spectrometric data and acquisition information were deposited to the ProteomeXchange Consortium (www.proteomexchange.org/) via the PRIDE partner repository (data set identifier: PXD013627).

## Code availability

All in-house developed computer code supporting the findings of this study are available from the corresponding author upon request.

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

## Acknowledgements

We acknowledge A. Leitner and R. Aebersold for sharing and maintaining their instrumentation. We are grateful to the members of the R.A. and B.W. research groups for suggestions and support at all stages of the project. We thank K. Novy for facilitating collaboration, J. R. Wyatt for text editing, J. Niebel and S. Sung for technical and T. Splettstoeser for graphical support. This work benefited from data assembled by the Immgen consortium. The following agencies are thanked for funding: ETH (grant ETH-30 17-1 and grant ETH-25 15-2) and Swiss National Science Foundation (grant 31003A_160259) for B.W. A.R. was holder of a chair in immunology endowed by L. Hoffman-La Roche Ltd, Basel.

## Author contributions

M.v.O. and M.M. performed all experiments except those noted below. F.K. and P.T. performed flow cytometry and FACS. M.v.O., M.M., R.B., L.R. and B.W. optimised and performed LC-MS/MS acquisition. M.v.O., M.M., F.K., P.T., R.B. and P.G.A.P. analysed data. P.G.A.P. performed glycosylation site analysis. H.Z. contributed new analytical tools. M.v.O., M.M., F.K., P.T., A.R. and B.W. designed research. M.v.O., M.M. and B.W. conceived the project and wrote and revised the paper.

## Competing interests

R.B. and L.R. are employees of Biognosys AG. Spectronaut is a trademark of Biognosys AG. All other authors declare no competing interests.
