## [Peer Review File · Nature Communications]

Reviewers' comments:

Reviewer #1 (Remarks to the Author):

In the manuscript by van Oostrum et al, the authors have developed an automated form of cell surface capture that has higher sensitivity and quantitative reproducibility relative to the original protocol developed in part by this same group. This innovation (i.e., automation) allows cell surfaceome analysis of cells that may be in limiting numbers (i.e., primary cell populations, among others). In developing this automated protocol, the authors show improved coefficient of variations in datasets relative to those generated by manual CSC protocols. These assessments were first performed on HeLa cells and then expanded to 11 common cancer cell lines.

Major claims made by the authors include an almost 5-fold increase in N-glycopeptide identification by MS using the autoCSC. Novel glycosylation sites, not previously reported, among the cancer cell lines were also identified, and these are listed in the tables. The relevance of these unique findings, as discussed in the paper, is limited, but identification of these sites is likely to be of relevance to the broader research field as this is an area of expanding and active investigation. Each of these findings represents a significant improvement over former CSC labelling technologies; however, these findings appear to be incremental. The analyses with the tumor cells seem to be an extension of the author's previous and at times published results.

Of more biological relevance was the application of this technology to B cell subtypes; however, the studies were only performed on mouse. The authors analyze a wide range of B-cell populations (ranging from Pre-B cells in the bone marrow to more mature B cells in the spleen and peritoneum) using the autoCSC. Of interest was the determination that CD20 preceded the presence of CD180. The former was first identified in the Large and Small Pre-B cells, while the later was first observed in the Immature B cells. The presence of CD20 in most/all of these cell types was already known; however, the distinctions with CD180 are of interest. These data were consistent with subsequent flow cytometry analyses and appeared to be useful in identifying different stages of B cells during their maturation process.

I find the improved protocols and results reported in this paper to be of interest to the research community. All experimental approaches and analyses (including statistics/bioinformatics) are aptly performed. There are no concerns.

The only real critique of the current study is the limited assessment of the mouse B cell populations. The technical advances are of significant value, but the biological scope and significance is more limited.. While the authors do identify 3 distinct populations (DN, SP and DP) for CD20 and CD180 in immature B cells, the description of BAFF-R differences and the differential presence of IgD on the surface of these cells is limited in scope.

- 1) Clearly the autoCSC has been used successfully with mouse B cells for proof of principle studies, but it would have been of broader interest to know if the same findings were relevant for human cells.
- 2) A greater focus on CD20 could also have broad appeal. This molecule is specifically a target for a number of monoclonal antibodies useful in clinic. As many/most of these antibodies target both mouse and human CD20, an assessment of the B-cell populations to these antibodies (and particularly the DN, SP and DP cells), would have strengthened the biological significant/impact of the current paper.
- 3) Alternatively, additional functional and/or molecular assessments of the populations described in Figure 2h,i and j may have been warranted. Potential differences in the IgD populations present in DN, SP and DP cells could also have been of interest. The presence of Ig molecules can be regulated at multiple levels, and how its presence is differentially regulated (if at all) between SP and DP cells could also be of interest.

Minor points:

- 1) Although the procedure described in this report is an automated process, it appears from the methodology section that there are aspiration steps through filter tips that are not automated. Please clarify these steps in greater detail.

2) Some of the superscripts were very small. This was apt for the references, but for the numbers, a larger font would have been useful

Reviewer #2 (Remarks to the Author):

The manuscript "De novo Classification of Mouse B Cell Types using Surfaceome Proteotype Maps" by van Oostrum and colleagues was informative and revealing of how fast advanced technologies are being applied to address simple but important questions. This is a straight-forward paper describing the miniaturization and automation of the previously described Cell Surface Capture technique first developed by Dr. Wollscheid and many of his colleagues on this paper. The bulk of the paper focuses on how the authors increased the sensitivity, reproducibility and throughput of this assay through their efforts. These efforts reveal the potential for this technology as there are many potential applications that this technology can be used to pursue.

Overall the paper was clear and the paper well written for a broad audience. Minor issues that may help others appreciate the finding are listed below.

1) The title of the paper is curious as the authors claim a de novo classification of mouse B cell types. However, the authors are not "starting from the beginning" as de novo implies since they are isolating B cell developmental subsets by FACS and then applying their technology to find additional potential markers for B cells at different stages. Thus, the authors may want to rethink their title. Further on this, the authors do not appear to describe how they have identified each of the B cell subsets that they focus on as subtle differences in methods/markers could generate different results. At a minimum, this needs to be described in detail since the paper relies on use of these de novo B cell subsets that have been defined over the past 30 years.

2) The authors should cite the reference paper for CSC at the point where it first appears in the manuscript, line 38, instead of line 45.

3) What is "SWATH-based" DIA?

4) Why is the automated and miniaturized method a more than 5-fold increase over the manual process (Fig. 1b)? It is unlikely to be because the humans carrying out the assay were inexperienced or careless. What factors made the difference? Similarly, what do the authors mean when they make the following statement and "To assess quantitative reproducibility, we performed MS1-based label-free quantification and calculated coefficients of variation (CVs) for precursors that match N-glycopeptides." What and how did the authors really do this as it is really difficult to understand what Fig. 1c is actually showing.

5) It is difficult to appreciate how valuable most of the data obtained are given that the authors quantified 1,697 unique glycosylated asparagines located within proteotypic peptides from 900 protein groups with a median of two sites per protein group (Supplementary Fig. 1, Supplementary Table 1). Since the median was two sites per protein group, this suggests that most of the data are binary, yes or no, but the data are not otherwise quantitative but become random and due to sampling errors, high or low. Perhaps the authors could expand on this concern.

6) It was difficult to discern the inter-assay variability, or did this reviewer just miss this? How good are the data if carried out at different times and then combined to discern differences? In part this question arises from the results described in issue #5 above.

7) The authors should clarify and correct their statement "Some proteins were found exclusively on a particular population, for example CD80 and CD130 on B1 cells (Supplementary Figure 5) and are potentially useful as stage-specific markers (Fig. 2b)." The B cell phenotype literature argues against this conclusion as these markers are not specific or limited to B1 cells, the authors assay may only be able to detect these molecules on B1 cells as the assay may not be sufficiently sensitive to detect these molecules on other B cells. In fact, the authors data support this reviewers conclusion as their data in supplemental figure 6 show that these molecules are not

found exclusively on a particular subset of B cells, but can be expressed at lower levels on other B cells.

8) As alluded in earlier comments, the authors may want to be more circumspect regarding some of their comments. For example, the authors state "Interestingly, flow cytometry revealed a bimodal distribution of CD20 within the immature B subpopulation (Supplementary Fig. 6). For CD180, we found a broad distribution covering more than two orders of magnitude (Supplementary Fig. 6)." This is not surprising as in part these results probably reflect how the authors are defining immature B cells as everyone in the field knows that "immaturity" is a subjective categorization. It would have helped had the authors told the reader how they were defining "immature B cells" in making this statement.

What is the basis for the authors statement that "We then asked whether the three subpopulations are different regarding maturity and found that the double-positive population had a significantly higher number of IgD+ cells (Fig. 2i) accompanied by higher surface abundance of BAFF receptor (Fig. 2j) compared to single-positive and double-negative subpopulations, indicating that the double positive subpopulation is at the verge of initiating migration from bone marrow to the spleen." This is speculation and should remain as such unless the authors have and present data to support this supposition.

9) In supplemental figure 5, the authors show similar levels of cell surface CD19 expression by all B cell subsets. However, this does not fit with most of the published FACS data for these B cell subsets, particularly for Pro-B cells. However, there was no description of how pro-B cells were defined, but they are generally published as being very low if not negative for CD19 expression in mice and there is a significant increase in CD19 expression density with B cell maturation. Perhaps the assay sensitivity is unable to detect these differences where are well represented in FACS plots.

As per the guidelines for the Journal, the brief nature of the paper and figure legends leaves many details of the paper unaddressed such as some nomenclature, figure axes and abbreviations not defined or explained, and the need for the reader to interpret what the figure is truly showing. The complexity of this problem increases as the reader digs deeper into the details of exactly what was done or is being shown. Given the space requirements, any effort to improve on this will be greatly appreciated by everyone who reads the paper.

Reviewers' comments:

Reviewer #1 (Remarks to the Author):

In the manuscript by van Oostrum et al, the authors have developed an automated form of cell surface capture that has higher sensitivity and quantitative reproducibility relative to the original protocol developed in part by this same group. This innovation (i.e., automation) allows cell surfaceome analysis of cells that may be in limiting numbers (i.e., primary cell populations, among others). In developing this automated protocol, the authors show improved coefficient of variations in datasets relative to those generated by manual CSC protocols. These assessments were first performed on HeLa cells and then expanded to 11 common cancer cell lines.

Major claims made by the authors include an almost 5-fold increase in N-glycopeptide identification by MS using the autoCSC. Novel glycosylation sites, not previously reported, among the cancer cell lines were also identified, and these are listed in the tables. The relevance of these unique findings, as discussed in the paper, is limited, but identification of these sites is likely to be of relevance to the broader research field as this is an area of expanding and active investigation.

In order to highlight the relevance of the identified glycosylation sites we performed additional analyses. The annotated sites in the UniProt database are largely based on computational prediction rather than experimental observation. Thus, we provide experimental evidence for 1017 human and 139 mouse glycosylation sites that were previously computationally predicted from sequence analysis. Supplementary Figure 2 has been updated and the following text added:

- *New Supplementary Fig. 2 panel c & d*

“However, 84% of the annotated sites in Uniprot are based on computational prediction, thus we provide experimental evidence for 1’017 Uniprot-predicted glycosylation sites.”

“Besides finding 25 new glycosylation sites, we provide experimental evidence for 139 computationally predicted sites. “

Each of these findings represents a significant improvement over former CSC labelling technologies; however, these findings appear to be incremental. The analyses with the tumor cells seem to be an extension of the author's previous and at times published results.

We sincerely appreciate the reviewer for recognizing the significant improvements achieved with autoCSC. However, the findings can hardly be described as incremental. A reduction of the input material down to the range of a few million cells enables for the first time comprehensive surfaceome analysis with cell types that were out of reach with former input requirement of 50-100 million cells. In combination with increased reproducibility and throughput, autoCSC now allows for large-scale phenotyping of cell populations of biomedical relevance and limited sample availability, which is the case for many primary cell types. Using autoCSC, we provide the first consistent surfaceome description across the hematological tree of B cells in mice which is the key model organism for immunologists.

The analysis with the cancer cell lines distinguishes itself from previous publications (Bausch-Fluck et. al. 2015) by the fact that all 11 cell lines were processed and directly quantified against each other in a single experiment. This was not possible in the past, the Cell Surface Protein Atlas from Bausch-Fluck et. al. 2015 describes an assortment of many different individually performed CSC experiments and is therefore limited in the quantitative comparison of surface proteins across cancer cell lines. Furthermore, the current analysis exceeds the median surface protein count per cell type two-fold, despite 5- to 33-fold lower input material.

Of more biological relevance was the application of this technology to B cell subtypes; however, the studies were only performed on mouse.

In order to demonstrate the applicability of autoCSC with primary cell types, we choose mouse B cell development because it is an important and extensively studied model system of developing immune cells. Furthermore, it is a particularly challenging model system for CSC, due to limited sample availability from mice, relatively small size, and little diversity in surfaceome composition. We believe that performing similar or same analysis on another model system would not add to the main message of the current manuscript, the surfaceome phenotyping of cell populations with limited sample availability at high throughput by autoCSC.

The authors analyze a wide range of B-cell populations (ranging from Pre-B cells in the bone marrow to more mature B cells in the spleen and peritoneum) using the autoCSC. Of interest was the determination that CD20 preceded the presence of CD180. The former was first identified in the Large and Small Pre-B cells, while the later was first observed in the Immature B cells. The presence of CD20 in most/all of these cell types was already known; however, the distinctions with CD180 are of interest. These data were consistent with subsequent flow cytometry analyses and appeared to be useful in identifying different stages of B cells during their maturation process.

I find the improved protocols and results reported in this paper to be of interest to the research community. All experimental approaches and analyses (including statistics/bioinformatics) are aptly performed. There are no concerns.

We sincerely appreciate the reviewer for recognizing that our findings are of interest and approval of experimental approaches and analysis.

The only real critique of the current study is the limited assessment of the mouse B cell populations. The technical advances are of significant value, but the biological scope and significance is more limited. While the authors do identify 3 distinct populations (DN, SP and DP) for CD20 and CD180 in immature B cells, the description of BAFF-R differences and the differential presence of IgD on the surface of these cells is limited in scope.

1) Clearly the autoCSC has been used successfully with mouse B cells for proof of principle studies, but it would have been of broader interest to know if the same findings were relevant for human cells.

We thank the reviewer for appreciating our study of mouse B cell development. We agree that translation of findings from mouse to human is generally of broad interest and importance. However, we believe that translational aspects are out of scope for this study and would distract attention from the key message of the current manuscript (see above).

2) A greater focus on CD20 could also have broad appeal. This molecule is specifically a target for a number of monoclonal antibodies useful in clinic. As many/most of these antibodies target both mouse and human CD20, an assessment of the B-cell populations to these antibodies (and particularly the DN, SP and DP cells), would have strengthened the biological significant/impact of the current paper.

We thank the reviewer for pointing out this interesting aspect. The identified immature B subpopulations are characterized by differential expression of CD20, and could therefore lead to differential susceptibility towards anti-CD20 antibodies within the immature B population. We decided to test this hypothesis and conducted additional experiments.

Interestingly, we found heterogeneity in sensitivity towards depleting anti-CD20 antibodies within the immature B cell population, in line with the identified subpopulations. Depletion was specific for the SP and DP subpopulations, while the DN immature B subpopulation even slightly increased. The findings are described in the new Figure 3 and the following text addition:

“Finally, we probed the immature B cell subpopulations in their sensitivity towards depleting anti-CD20 antibodies. In agreement with CD20 expression levels, SP and DP subpopulations were depleted upon anti-CD20 antibody treatment, while the DN cells even slightly increased in counts (Supplementary Fig. 8a,b).”

3) Alternatively, additional functional and/or molecular assessments of the populations described in Figure 2h,i and j may have been warranted. Potential differences in the IgD populations present in DN, SP and DP cells could also have been of interest. The presence of Ig molecules can be regulated at multiple levels, and how its presence is differentially regulated (if at all) between SP and DP cells could also be of interest.

We thank the reviewer for this suggestion. We performed additional experiments for further functional and molecular assessment of the immature B subpopulations DN, SP and DP:

- We assessed the developmental trajectory of DN, SP and DP cells in culture and validated that both DN and SP develop into DP cells in vitro. These findings provide direct evidence for the implied directionality of development from DN to DP.

“We then asked whether the three subpopulations follow a developmental trajectory towards maturity. To test the hypothesis that the DN population differentiates to DP, we cultured CD19+IgM+CD93+ DN, SP and DP cells in vitro while monitoring CD20 and CD180 surface expression. In line with our hypothesis, we found that both DN and SP populations develop into DP within 3 days of culturing (Fig. 3b).”

- We also performed a functional analysis of the identified immature B sub-populations by assessing their response to BAFF in vitro. As shown in Fig. 3, BAFF significantly increased the in vitro survival of only the DP sub-population, in agreement with the expression pattern of BAFF receptor within the three sub-populations.

“Based on these findings, we assessed the identified immature B subpopulations in their response to BAFF in vitro. Interestingly, BAFF treatment significantly increased in vitro survival of only the DP subpopulation (Fig. 3e), in agreement with the higher expression pattern observed for DP.”

- We performed further proteotype analysis of the identified immature B subpopulations and found that the three immature B subpopulations are distinct on proteotype level reflecting the phenotypic classification.
 - **Supplementary table 3**

“For molecular assessment of the proteotypes underlying the observed phenotypes, we performed quantitative proteomic analysis of the identified immature B subpopulations (Supplementary Table 3). Also, on the proteotype level, we found distinct clustering of DN, SP and DP subpopulations in a principal component analysis (Fig. 3f) and hierarchical clustering of protein quantities (Supplementary Fig. 7). Comparing all three subpopulations, we identified 435 proteins with significantly modulated

abundance (Supplementary Table 3), including proteins involved in B-cell receptor signaling (e.g. CD79A/B, IgM) and antigen presentation (e.g. HB2A, HG2A) (Supplementary Fig. 7b-c).”

Minor points:

1) Although the procedure described in this report is an automated process, it appears from the methodology section that there are aspiration steps through filter tips that are not automated. Please clarify these steps in greater detail.

All aspiration steps through filter tips are fully automated by the liquid handling robot. We clarified this point in the method section.

2) Some of the superscripts were very small. This was apt for the references, but for the numbers, a larger font would have been useful

We increased the font size in the manuscript.

Reviewer #2 (Remarks to the Author):

The manuscript “De novo Classification of Mouse B Cell Types using Surfaceome Proteotype Maps” by van Oostrum and colleagues was informative and revealing of how fast advanced technologies are being applied to address simple but important questions. This is a straight-forward paper describing the miniaturization and automation of the previously described Cell Surface Capture technique first developed by Dr. Wollscheid and many of his colleagues on this paper. The bulk of the paper focuses on how the authors increased the sensitivity, reproducibility and throughput of this assay through their efforts. These efforts reveal the potential for this technology as there are many potential applications that this technology can be used to pursue.

Overall the paper was clear and the paper well written for a broad audience. Minor issues that may help others appreciate the finding are listed below.

We thank the reviewer for appreciation of our study, we addressed all raised points below.

1) The title of the paper is curious as the authors claim a de novo classification of mouse B cell types. However, the authors are not “starting from the beginning” as de novo implies since they are isolating B cell developmental subsets by FACS and then applying their technology to find additional potential markers for B cells at different stages. Thus, the authors may want to rethink their title.

We thank the reviewer for this suggestion and changed the title accordingly.

- “Classification of mouse B cell types using surfaceome proteotype maps”

Further on this, the authors do not appear to describe how they have identified each of the B cell subsets that they focus on as subtle differences in methods/markers could generate different results. At a minimum, this needs to be described in detail since the paper relies on use of these de novo B cell subsets that have been defined over the past 30 years.

The sorting strategy is illustrated in detail in Supplementary Fig. 4.

We additionally added the sorting details to the methods section.

2) The authors should cite the reference paper for CSC at the point where it first appears in the manuscript, line 38, instead of line 45.

We thank the reviewer for noticing and added the reference at line 38.

3) What is “SWATH-based” DIA?

The term SWATH was introduced in Gillet et. al 2012 (Ref. 13) and refers to a concept for proteome analysis by targeted feature extraction of MS/MS spectra generated by data independent acquisition. We changed the phrasing and position of the reference in the main text for clarification.

“Targeted feature extraction and data-independent acquisition (DIA)(Gillet et al. 2012) are used to quantify surface-protein abundances across multiple conditions.”

4) Why is the automated and miniaturized method a more than 5-fold increase over the manual process (Fig. 1b)? It is unlikely to be because the humans carrying out the assay were inexperienced or careless. What factors made the difference?

We thank the reviewer for this interesting consideration. While we don’t know in detail why the automated and miniaturized method outperforms manual processing, we hypothesize the following factors mediate the difference:

- No loss of beads, no transferring of beads between reaction tubes
- Decreased surface area of plastics that would lead to nonspecific retention of peptides
- Beneficial kinetics within the confined reaction space

Similarly, what do the authors mean when they make the following statement and “To assess quantitative reproducibility, we performed MS1-based label-free quantification and calculated coefficients of variation (CVs) for precursors that match N-glycopeptides.” What and how did the authors really do this as it is really difficult to understand what Fig. 1c is actually showing.

With the hope to improve clarity, we integrated the following passage into the main text:

“Next to the 5-fold increased sensitivity, we further asked whether autoCSC allows for more reproducible quantification of cell surface derived N-glycopeptides compared to the manual workflow. For this, deamidated N-glycopeptides were label-free quantified by integration of mass spectrometry monitored chromatographic traces. For every N-glycopeptide, quantitative values across ten replicates were used to calculate a coefficient of variation (CV) for autoCSC, experimenter 1 and experimenter 2 as measure for its quantitative reproducibility. The overall distribution of obtained peptide CVs per designated workflow are visualized in a box plot in Fig. 1c.”

5) It is difficult to appreciate how valuable most of the data obtained are given that the authors quantified 1,697 unique glycosylated asparagines located within proteotypic peptides from 900 protein groups with a median of two sites per protein group (Supplementary Fig. 1, Supplementary Table 1). Since the median was two sites per protein group, this suggests that most of the data are binary, yes or no, but the data are not otherwise quantitative but become random and due to sampling errors, high or low. Perhaps the authors could expand on this concern.

The peptides resulting from autoCSC experiments are subjected to LC-MS/MS analysis and abundance quantification using the described DIA approach (Gillet. et al. 2013). Therefore, we obtain quantitative (not binary) values for peptides that can be used for relative quantification across conditions (data in supplementary tables).

As depicted in Fig. 1a, in autoCSC, extracellular glycopeptides are tagged, enriched and released using PNGase F, leaving a deamidation motif at asparagines within the glycosylation motif (NXS/T). Therefore, we can identify initially labelled and extracellularly exposed peptides with the specific glycosylation site by LC-MS/MS analysis if they are (a) proteotypic and (b) contain a deamidated asparagine (c) within the NXS/T glycosylation motif.

One unique glycosylation site can be present on different peptides (due to e.g. different charge states, missed cleavages by trypsin, or multiple glycosylation site on one peptide etc.), therefore we used bioinformatic analysis to report the number of unique glycosylation sites identified and from which protein groups they originate. In the median, we identify two unique glycosylation sites for each protein group (the distribution of site per protein groups is depicted in Supplementary Fig. 1).

6) It was difficult to discern the inter-assay variability, or did this reviewer just miss this? How good are the data if carried out at different times and then combined to discern differences? In part this question arises from the results described in issue #5 above.

autoCSC is a spatial proteotype analysis workflow and relies on LC-MS/MS based quantitative proteomics as readout. As such, the same principles and limitations of LC-MS/MS based protein

quantification apply here as well. The LC-MS/MS based quantification method used only allows for relative quantification (as opposed to absolute quantification) across different samples which were processed and measured simultaneously. This is due to parameters related to LC-MS/MS acquisition, such as the particular performance of the mass spectrometer or chromatographic setup at the time of measurement, but also processing steps such as efficiency of enzymatic digestion. These considerations are not specific for autoCSC, but valid for LC-MS/MS-based quantitative proteomics in general.

Therefore, it is desirable to have a workflow that allows for simultaneous processing of multiple samples. autoCSC bridges this gap and enables parallel processing and analysis of independent biological replicates resulting in reproducible cell surface protein quantification (median CV of 28%). The high-throughput capabilities of autoCSC allow for simultaneous processing of up to 96 samples and thereby circumvents the aforementioned sources of variance in large-scale surfaceome experiments.

7) The authors should clarify and correct their statement “Some proteins were found exclusively on a particular population, for example CD80 and CD130 on B1 cells (Supplementary Figure 5) and are potentially useful as stage-specific markers (Fig. 2b).” The B cell phenotype literature argues against this conclusion as these markers are not specific or limited to B1 cells, the authors assay may only be able to detect these molecules on B1 cells as the assay may not be sufficiently sensitive to detect these molecules on other B cells. In fact, the authors data support this reviewers conclusion as their data in supplemental figure 6 show that these molecules are not found exclusively on a particular subset of B cells, but can be expressed at lower levels on other B cells.

We thank the reviewer for noticing and adjusted the text accordingly.

“Few proteins we found exclusively on a particular population, for example CD80 and CD130 on B1 cells.”

8) As alluded in earlier comments, the authors may want to be more circumspect regarding some of their comments. For example, the authors state “Interestingly, flow cytometry revealed a bimodal distribution of CD20 within the immature B subpopulation (Supplementary Fig. 6). For CD180, we found a broad distribution covering more than two orders of magnitude (Supplementary Fig. 6).” This is not surprising as in part these results probably reflect how the authors are defining immature B cells as everyone in the field knows that “immaturity” is a subjective categorization. It would have helped had the authors told the reader how they were defining “immature B cells” in making this statement.

Immature B cells are defined as in suppl. Fig. 4 sorting strategy, and now additionally also in the methods section. We call them “immature B” in order to distinguish them from re-circulating mature B cells, which are also IgM+ but CD93-. In contrast to re-circulating mature B cells, immature B have not yet migrated to the spleen where they complete their maturation. They are also clearly distinguished from earlier pro-B and pre-B cells since they have completed Ig rearrangements and express IgM on their surface.

What is the basis for the authors statement that “We then asked whether the three subpopulations are different regarding maturity and found that the double-positive population had a significantly higher number of IgD+ cells (Fig. 2i) accompanied by higher surface abundance of BAFF receptor (Fig.

2j) compared to single-positive and double-negative subpopulations, indicating that the double positive subpopulation is at the verge of initiating migration from bone marrow to the spleen.” This is speculation and should remain as such unless the authors have and present data to support this supposition.

We thank the reviewer for pointing this out. We removed the sentence and now present data to support the implied developmental trajectory (the in vitro differentiation DNSPDP, Fig. 3b)

“We then asked whether the three subpopulations follow a developmental trajectory towards maturity. To test the hypothesis that the DN population differentiates to DP, we cultured CD19+IgM+CD93+ DN, SP and DP cells in vitro while monitoring CD20 and CD180 surface levels. In line with our hypothesis, we found that both DN and SP populations develop into DP within 3 days of culturing (Fig. 3b).”

9) In supplemental figure 5, the authors show similar levels of cell surface CD19 expression by all B cell subsets. However, this does not fit with most of the published FACS data for these B cell subsets, particularly for Pro-B cells. However, there was no description of how pro-B cells were defined, but they are generally published as being very low if not negative for CD19 expression in mice and there is a significant increase in CD19 expression density with B cell maturation. Perhaps the assay sensitivity is unable to detect these differences where are well represented in FACS plots.

B cell developmental stages have been named differently by different labs. In the present manuscript, we define pro-B cells as CD19+CD117+IgM-. We would call CD19-B220+ cells as “pre-pro-B cells”. These cells are not included in our analysis.

Supplementary Fig. 6 shows FACS data for CD19 expression on the B cell populations used in this manuscript.

As per the guidelines for the Journal, the brief nature of the paper and figure legends leaves many details of the paper unaddressed such as some nomenclature, figure axes and abbreviations not defined or explained, and the need for the reader to interpret what the figure is truly showing. The complexity of this problem increases as the reader digs deeper into the details of exactly what was done or is being shown. Given the space requirements, any effort to improve on this will be greatly appreciated by everyone who reads the paper.

We added explanations for nomenclature, figure axes and abbreviations throughout the manuscript for clarity and compliance with the editorial policy.

REVIEWERS' COMMENTS:

Reviewer #1 (Remarks to the Author):

The authors have submitted a revised manuscript with a number of clarifications, the addition of new experiments and improved data analysis.

First, I thank the authors for the clarification regarding my incremental improvement. I had read a number of papers where authors were using substantially less than the 50-100 million cells required for the standard cell surface capture methodology. Clearly 30 million or less cells could be used, but you are correct in stating that the numbers employed here are significantly reduced and are apt for limiting populations like B cells.

Second, the addition of data regarding the identified cell populations (and the new title focused on mouse B cells) have improved my view of the overall impact of this paper. It is unfortunate that no human data will be added regarding the markers you identified; however, the additional studies are welcome and I believe that this has improved the overall relevance of this paper, particularly since it has now added some novel findings that have clearly come from the output of your improved methodology.

Reviewer #2 (Remarks to the Author):

None.

REVIEWERS' COMMENTS:

Reviewer #1 (Remarks to the Author):

The authors have submitted a revised manuscript with a number of clarifications, the addition of new experiments and improved data analysis.

First, I thank the authors for the clarification regarding my incremental improvement. I had read a number of papers where authors were using substantially less than the 50-100 million cells required for the standard cell surface capture methodology. Clearly 30 million or less cells could be used, but you are correct in stating that the numbers employed here are significantly reduced and are apt for limiting populations like B cells.

Second, the addition of data regarding the identified cell populations (and the new title focused on mouse B cells) have improved my view of the overall impact of this paper. It is unfortunate that no human data will be added regarding the markers you identified; however, the additional studies are welcome and I believe that this has improved the overall relevance of this paper, particularly since it has now added some novel findings that have clearly come from the output of your improved methodology.

Reviewer #2 (Remarks to the Author):

None.

We thank both reviewers for their valuable comments and suggestions.